# Shift in Circulating Serum Protein Fraction (SPF) Levels of Pregnant Jennies and Nutritional Related Aspects at Early-, Mid- and Late Gestation

**DOI:** 10.3390/ani11092646

**Published:** 2021-09-09

**Authors:** Maria Grazia Cappai, Petra Wolf, Annette Liesegang, Giovanni Paolo Biggio, Andrea Podda, Antonio Varcasia, Claudia Tamponi, Fiammetta Berlinguer, Ignazio Cossu, Walter Pinna, Raffaele Cherchi

**Affiliations:** 1Dipartimento di Medicina Veterinaria, University of Sassari, 07100 Sassari, Italy; andreapodda94@tiscali.it (A.P.); varcasia@uniss.it (A.V.); cltamponi@uniss.it (C.T.); berling@uniss.it (F.B.); prodanim@uniss.it (W.P.); 2Institute of Nutrition Physiology and Animal Nutrition, University of Rostock, Justus-von-Liebig-Weg 6b, 18059 Rostock, Germany; Petra.Wolf@uni-rostock.de; 3Institute of Animal Nutrition, Vetsuisse Faculty, University of Zurich, Winterthurerstr. 270, 8057 Zurich, Switzerland; aliese@nutrivet.uzh.ch; 4Dipartimento di Ricerca Per gli Equini, Agris Sardegna, 07014 Ozieri, Italy; giampaolo.biggio@tiscali.it (G.P.B.); icossu@agrisricerca.it (I.C.); ilex283@gmail.com (R.C.)

**Keywords:** albumin, donkey, fetus, globulins, pregnancy

## Abstract

**Simple Summary:**

The nutritional management of gestating animals requires the adequate provision of nutrients and energy through the diet and an appropriate drinking water supply. While those are the basic principles to ensure proper feeding of pregnant animals, the fate of nutrients in the post absorption stage, with consequent distribution and delivery to key organs, appears to be pivotal for their optimal utilization. Besides, the augmented nutrient and energy requirements of the mother/fetus binomial justify the monitoring of metabolic conditions, crucial for correct nutritional assessment. In this regard, levels of serum protein fractions (SPF) circulating in the bloodstream of gestating animals are known to shift across the different phases of pregnancy in several animal species, including in women. In fact, some of these serum proteins serve as carriers for a series of molecules of strategic importance in the metabolic crosstalk between the mother and the developing fetus. The literature regarding SPF levels in pregnant jennies appears to be limited at present. Results from the monitoring of SPF in pregnant Sardo breed jennies at early-, mid-, and late gestation are reported here.

**Abstract:**

A viable tool for the monitoring of the systemic condition of the pregnant jenny may be the determination of serum protein fraction (SPF) levels, including metabolic profiling. Tissue development and composition of the growing fetus requires the mother to provide adequate nutrients to its body parts and organs. In this regard, body fluid distribution and strategic molecule transportation can be screened using SPF electropherograms and analysis of intermediate metabolites. The nutritional and health status of 12 jennies (age: 5–8 years; BW at the start: 135–138 kg; Body Condition Score, BCS [1 to 5 points] = 2.25–2.50; 4th month of gestation) were monitored throughout gestation (approximate gestation period 350–356 d). All animals were pasture-fed and were offered hay *ad libitum*. Individual blood samples were collected within the 4th, 7th, and 10th month following conception (ultrasound scanning). Serum biochemistry, in particular, the analysis of 6 fractions of serum proteins was carried out. The significant decrease in circulating albumin in jennies from mid- to late-gestation (*p* < 0.001) suggests a considerable role of dietary amino acids in the synthesis of protein for fetal tissue formation as well as body fluid distribution and blood pressure control of the jenny in those stages. Moreover, α_1_-globulin decreased significantly in late gestation (*p* < 0.047), corresponding to major organ development in the terminal fetus and supported by lipid transportation in the bloodstream of the jenny. Similarly, α_2_-globulin decreased in late gestation (*p* < 0.054) as haptoglobin, an important component for the transport of free circulating hemoglobin, is likely used for fetal synthesis. Mid-gestation, appears to be a crucial moment for adequate dietary nutrient supplementation in order to prevent homeostasis perturbation of jennies, as observed in this trial.

## 1. Introduction

Adequate feeding of the pregnant jennies appears to be crucial in donkey farming. Gestation represents a very complex metabolic process for mammals and consequently a hard task to manage for the clinical nutritionist. The interplay between mother and fetus requires the need to monitor the nourishment and health condition of both. In particular, the regulating role of developing tissues and organs of the fetus across the different phases of pregnancy on the metabolic activity of the mother [1,2].

However, the physiological adaptation to gestational phases involves a series of changes also for the mother, of increasing extent as long as pregnancy proceeds and foaling approaches. In this metabolic scenario, the nutritional management of gestating animals necessarily involves the adequate provision of nutrients and energy with the diet, along with drinking water supply. In general terms, nutrient utilization and body fluid distribution undergo marked systemic rearrangements, above all if the post absorption fate of dietary key-molecules is considered. Against this background, endogenous synthesis and distribution of nutrients and related molecules appear to be pivotal to ensure the health and nourishment of the mother/fetus binomial.

A viable tool to ensure adequate monitoring of the systemic condition of the pregnant jenny and growing fetus may be the determination of serum protein fractions (SPF) [3]. Body fluid distribution and its respective role in the systemic nutrient delivery to body parts and organs can in part be screened by SPF electropherograms. Levels of serum protein fractions (SPF) circulating in the bloodstream of gestating animals are known to shift across different phases of pregnancy in several animal species [3] including the pregnant woman [4]. We, therefore, hypothesized that SPF levels could have a prognostic value in the pregnant jenny as well.

The aim of the present investigation was to assess the circulating levels of SPF in pregnant jennies at early-, mid-, and at late gestation and to describe the metabolic environment across these different stages in order to explore their role in the nutritional assessment of the mother/fetus binomial in view of this particular physiological condition.

## 2. Materials and Methods

### 2.1. Animal Care

Animal handling complied with the recommendations of the European Union Directive 2010/63/EU concerning animal care. All procedures reported in this trial were carried out by professionals according to conventional clinical practices; in particular, blood sampling was carried out by expert veterinary practitioners and animals were handled by trained technicians. The methods described align to the ARRIVE guidelines. The experimental activities were approved by the Ethics Committee of the University of Sassari (n. 110672, 31/08/2021).

### 2.2. Animal Feeding and Management Conditions

A total of 12 Sardo breed jennies were enrolled in the trial. All animals belonged to the Autonomous Region of Sardinia (RAS), born and raised at the Department of Research for Equine Breeding and Production (DREBP) of Agris (Agency of Research of the RAS). The age of the animals ranged from 5 to 8 years, with a bodyweight at first check of 135–138 kg and a Body Condition Score [BCS range, 1 to 5 points] of 2.25–2.50. All jennies underwent breeding service at the DREBP, following the Regional regulations for equine breeding and reproduction (Law 30/1991; DGR 4/35/1995; DM 403/2000). Upon ultrasound scanning for successful conception, the nutritional status of all jennies was assessed following the approach by Cappai and co-workers [5] developed for the Sardinian donkey. The approximate gestation period of the Sardo donkey is of 350–356 days and health screenings were scheduled following the internal protocols of the DREBP; at early-, mid-, and late gestation. Thus, each jenny was physically assessed within the fourth, seventh, and tenth month of gestation. During each control, all jennies underwent nutritional assessment following the same approach as mentioned before [5].

The feeding regime of the animals changed substantially along the trial due to the length of the investigation (because of seasonal changes). All jennies enrolled were kept and fed consistently on open pasture (whilst sheltered during the nighttime) according to feeding practices developed for donkeys at the DREBP. To exceed the amount of 1.5% of BW fed, jennies were offered hay in addition to pasture. Following, feed offer was increased, estimating a daily dry matter intake of 1.8% of BW, when animals entered the last third of gestation. Over the entire trial, independent of the season, free access to barley straw was provided through feeders placed directly on pastures. Finally, when sheltered, jennies were kept in individual boxes with straw litter, expected to be consumed as well and, by late gestation, 0.10 kg of compound feed (mixed concentrate feed) for horses at maintenance was provided in the box. However, the last two months of gestation fell out of the scheduled checks (until the 10th month of gestation) in this trial. A summary of the rations provided to the animals as well as the chemical composition of the fed hay is reported in Table 1.

### 2.3. Blood Analysis

Blood samples were taken at each assessment following the internal protocols applied at the DREBP. Sampling occurred at 8:00 a.m., 2 h after hay administration. Blood was drawn directly into vacuum tubes through the puncture (18 gauge needle) of the jugular vein. All samples were identified with the jenny’s name, electronic individual code (EIC), and date of sampling. One aliquot of the blood of each animal was stored and transported to the lab for biochemical profiling. Samples underwent centrifugation at 1500× *g* for 10 min and serum was removed and stored in a vial (2 mL) to be frozen at −20 °C until analysis. All samples were analyzed within one week using an automatic biochemical analyzer (Mindray BS-200, Alcyon, Italy) for the determination and quantification of hepatic enzymes (alanine transaminase, ALT; aspartate aminotransferase, AST; gamma-glutamyl transferase, γ-GT) and other biochemical parameters (urea; total bilirubine, TBil; total protein, TP; total cholesterol, TC; total triglycerides, TG; glucose, GLU; lactate dehydrogenase, creatinine, CREA; alkaline phosphatase, ALP; calcium, Ca; phosphorus, P; iron, Fe). Reference intervals were adjusted for donkey species and biochemical parameters obtained were compared to those reported by Caldin and co-authors [8].

Serum protein fraction electrophoresis was carried out through fully automated equipment (Pretty Interlab^®^, Rome, Italy). From each sample, 30 μL of serum was used for a six protein run to determine albumin and globulins (α_1_-, α_2_-, β_1_-, β_2_-, γ-globulins) concentrations in serum, expressed as g/L and as a percentage (%) of serum total protein concentration (g/L).

### 2.4. Analysis of Data and Statistics

Body weight (BW) was recorded for each jenny on a digital scale at each assessment.

The chemical composition of hay and straw was used to estimate the digestible energy (DE) content per kg of dry matter (DM) in feed. Calculations were based on the following formula [9]:

Forages:DE (Mcal/kg DM)=2.118+0.01218CP−0.00937ADF−0.0383(NDF−ADF)+0.04718EE+0.02035NFC−0.0262Ash,

The statistical analysis of data was carried out through the application of ANOVA in a mixed procedure model:Y_i,j,k_ = μ + D_i,j,k_ + H + e_i,j,l,k_
where Y is the dependent variable (CBC parameters; biochemical parameters), μ is the overall mean and D is the fixed effect of the sampling time (three levels: 4 mo.- vs. 7 mo.- vs. 10 mo. of gestation). Animal (H) represents the random effect and e is the random error.

Confidence intervals and groupings were adjusted according to Tukey’s method. All data were analyzed using SAS 9.2 (SAS Inst. Inc., Cary, NC, USA). Statistical significance was set for *p*-value < 0.05, whereas *p*-value < 0.10 represented a trend.

Selected metabolites were correlated with each other to depict liver function and renal clearance of the gestating jennies. Pearson’s test for the assessment of potential correlation (ρ < 0.300 = weak correlation; 0.300 < ρ < 0.600 = mild correlation; 0.600 < ρ < 1.000 strong correlation; +ρ or –ρ, positively or negatively correlated, respectively; significance for *p*-value < 0.005) was used.

## 3. Results

All animals enrolled in the trial appeared healthy throughout the experimental period. Jennies showed an increasing BW from 4 mo. on, achieving a BW = 143 ± 2.31 kg at 7 mo. (Differential, Δ = +4.37%) and 157 ± 2.15 kg (Δ = +14.6%). Jennies appeared lean and were scored a BCS of 2.25–2.50 throughout the trial. The biochemical profile of selected metabolites of jennies at different phases of gestation is reported in Table 2.

Metabolic profiles varied according to the time of gestation for some parameters. In particular, while being above the upper limit of the reference range in serum at the first and second blood sampling, TP significantly decreased (*p* = 0.031) with increasing gestation. Similarly, although no significant difference was found over time (*p* = 0.061), UREA levels exceeded the upper value of the reference range in jennies at 4th mo. and decrease progressively at 7th mo. and 10th mo. of gestation. Serum iron was found to decrease progressively as well, though with huge variation across sampling times and with no significant difference.

Liver activity (albumin) and function (TB, UREA, ALT and AST) and renal clearance (CREA, UREA) in relation to body fluid distribution were screened to evaluate their correlation across the three different gestation phases. Table 3 shows the correlation coefficients (ρ) and statistical significance (*p*-value) of selected parameters.

TP was shown to be positively correlated with ALB (ρ = 0.445; *p* = 0.020), AST (ρ = 0.595; *p* = 0.001) and UREA (ρ = 0.579; *p* = 0.002) in a moderate way. Moreover, a negative correlation between TP and circulating CREA (ρ = −0.540; *p* = 0.004), which in turn markedly correlated with ALT and moderately with UREA in a negative and highly significant way [(ρ = −0.703; *p* < 0.0001) and (ρ = −0.560; *p* < 0.0001), respectively] was found. As a trend, while CREA and TB increased progressively, UREA, ALB, and ALT decreased as gestation proceeded. Taken altogether, these parameters have both nutritional and additional value in terms of homeostasis maintenance.

Values of circulating levels of SPF across the gestation phases is reported in Table 4.

Albumin levels decreased significantly from mid- to late-gestation (*p* = 0.001). Similarly, α_1_-globulins decreased significantly during late gestation (*p* = 0.047), corresponding to massive organ development in the terminal fetus and supported by lipid transportation in the bloodstream of the jenny. Though non significantly, α_2_-globulins decreased during late gestation as haptoglobin serves as an important component for the transport of free circulating hemoglobin, entering the iron cycling in the spleen. γ-globulins decreased from the 4th mo. progressively to the 7th and the 10th mo. of gestation (*p* = 0.037).

## 4. Discussion

The monitoring of the metabolic profile of gestating jennies corresponding to the evolution of nutrient distribution in view of the systemic demand of the mother/fetus binomial over time was found to be an efficacious tool. In particular, changes in SPF values shed new light on the physiological meaning and nutritional role of the circulating values of these compounds, yet explored in the jenny. Adaptations of the pregnant jenny according to increased metabolic activity and development of fetal tissues as well as global adaptations of the reproductive organs and annexes along the gestation period could be obviously expected. However, the monitoring of the evolution of such metabolic rearrangements in the jenny is novel and pivotal for decision making when nutritional practices are involved. Thus, in view of the significant shift of different circulating metabolites, the overall condition of the pregnant jennies enrolled in this trial could be described.

In this regard, it is useful to state that comparative investigations about digestion physiology and feed utilization in horses and donkeys have revealed nutritional requirements distinctive for each animal species [10,11,12,13,14,15]. In the light of such knowledge, feeding the pregnant jenny may turn out to be different from the feeding practices of the mare during gestation. Nowadays, the nutrition and feeding of donkeys are well distinguished from nutritional assessments and feeding practices in horses [16,17,18,19,20,21,22,23]. Nevertheless, despite the recent interest of the scientific community, the present knowledge of donkey nutrition appears limited and gaps remain in various topics, deserving better understanding [24,25].

Consolidated feeding practices of the DREBP allowed pregnant jennies to have a feed offer exceeding their daily DM intake capacity (as a percentage of BW, exceeding 1.8% in the last period). Furthermore, gestating animals were given the chance to select feed stuffs, both when on pasture and sheltered in the box. Regardless, some shifts in circulating values during gestation are not limited to nutritional aspects (UREA, ALB, α_1_- and γ-globulins, with average values showing to vary with a *p* < 0.05). In fact, variations in some metabolites can be acknowledged to other homeostatic processes, beyond tissue nourishment and energy status, such as regulation of blood pressure and organ functions (mainly liver and kidneys). Such adaptations thus do contribute to varying metabolic patterns over time and across the gestation stages.

Due to the lack of additional data on SPF levels in pregnant jennies throughout gestation, results obtained in this trial were interpreted in view of the general physiological and metabolic information available in the literature [21,22,23,24,25,26,27,28,29,30,31,32].

One first aspect concerns the importance of metabolic screening at different stages of gestation (adequate timing). As a practical recommendation, due to an expectable decrease in daily DM intake at late gestation [23], the last third of gestation is considered optimal for the provision of additional amounts of energy through high-quality feed supplementation in donkeys. In view of the metabolic shifts, represented by the changes in circulating parameters observed in this trial, it appears that for pregnant Sardo breed jennies such changes in feeding management should be implemented at the 7th mo. of gestation (mid gestation).

The diagnostic role of selected metabolites investigated in this trial includes several aspects beyond nutritional purposes within the gestating jenny. Liver and kidney functions could be evaluated and appeared to be conserved at each check. The values obtained did not show any prognostic indication of organ failure. Nevertheless, the role of protein metabolism during gestation in the jenny deserves further in-depth consideration.

ALT and AST are enzymes involved in the deamination and transamination of amino acids for the production of energy, both of endogenous (tissue catabolism) and exogenous (dietary and derivates) origin. The final side product produced during this process is ammonia, converted by the liver into the nontoxic form UREA, which is easily excreted. UREA, however, is not solely an excretion product but is also involved in body fluid distribution crucial for blood pressure maintenance in the gestating animal. CREA values in serum mirror kidney function and clearance, which circulating levels in our investigation, tended to increase as gestation proceeded (UREA tended to decrease over time). ALB, besides serving as a carrier for various molecules [21,22], represents an efficient homeostasis-keeping tool as well as it is also involved in body fluid distribution. ALB decreased significantly over time in the bloodstream of gestating jennies, consistent with what is reported for women [4]. Likewise, α_1_- (*p* = 0.047) and α_2_-globulins (*p* = 0.052) decreased over time and proportionally contributed in lowering the ALB to globulin ratio. The meaning of fluctuating levels of α-globulins can therefore be interpreted differently. As carriers, α_1_-globulins are mainly involved in the transport of various molecules of metabolic importance, fundamental for energy purposes. For instance, α_1_-globulins function as lipid transporters, as energy substrates, and as integral molecules of the cell membrane for the developing tissues of the fetus. In the same way, α_2_-globulins serves as a carrier as well and binding molecules for a series of different proteins, including free hemoglobin (similarly to the function of haptoglobin). Acute-phase proteins were not considered in this research. As all jennies appeared healthy, and normal values for TB were recorded throughout the trial, no hemolytic effect could be assessed. The levels of β-globulins, of which transferrin is the most abundant protein (involved in the transport of iron), were relatively constant throughout the trial. However, iron decreased progressively (non-significantly, *p* = 0.570) over time even though staying above the upper value of the reference range.

## 5. Conclusions

The analysis of serum protein fractions and metabolic profiles from selected metabolites in this research demonstrated the importance of monitoring the pregnant jenny across the different stages of gestation. The shift in parameters investigated here revealed physiological and nutritional variations at the base of the metabolic rearrangement within the mother during fetal development. Exploration of liver and kidney function, together with the SPF, highlighted how sharply augmented requirements over time can shape the systemic distribution of molecules to various tissues. As a result, the proper moment to adapt nutritional practices (high-quality feed supplementation) in Sardo breed donkeys was determined to be in the mid-term of gestation (7th month).

## Figures and Tables

**Table 1 animals-11-02646-t001:** Rations for the pregnant jennies across different stages of gestation.

Item ^2^	Gestation ^1^		
	4 mo.	7 mo.	10 mo.
Ingredient (kg/d per jenny, as fed)			
Hay ^3^	0.90	0.90	0.90
Straw ^4^	0.90	0.90	0.90
Pasture	+	+	+
Chemical composition of hay (g/kg feed)			
DM	847		
NDF	585		
CP	129		
Ash	135		
Ether extract	21.6		
DE (Mcal/kg DM in feed) ^5^	2.44		

^1^ Gestation: 4-mo. = 4th month of gestation; 7-mo. = 7th month of gestation; 10-mo. = 10th month of gestation. ^2^ Item: DM, dry matter; NDF, Neutral Detergent Fiber; CP, crude protein; ^3^ Hay mown from homegrown mixed oat/ryegrass/clover sewed over two years. ^4^ Barley straw, chemical composition (%): DM = 92%; CP = 4.9; NDF = 81.3%, DE (Mcal/kg DM) = 1.91. ^5^ DE = digestible energy per kg of dry matter in feed expressed as Mega calories. Energy density for concentrate and forage was predicted by the formula of Pagan and Hintz (1986) [6] and NRC, 2007 [7].

**Table 2 animals-11-02646-t002:** Metabolic profile of jennies across the different gestation phases.

		Gestation ^1^				*p*-Value ^4^	
Item ^2^	Reference Range ^3^	4-mo.	7-mo.	10-mo.	Pooled-SD	Sampling Time	Animal
TP (g/L)	62.0–75.0	88.0	74.5	73.2	5.468	0.031	0.468
UREA (mg/dL)	11.0–39.0	48.6	33.3	33.1	8.149	0.061	0.879
ALT (U/L)	5.0–14.0	9.32	6.97	7.13	6. 228	0.855	0.650
AST (U/L)	279–430	389	271	239	67.05	0.517	0.029
TB (mg/dL)	0.07–0.21	0.16	0.17	0.20	0.008	0.592	0.650
Ca (mmol/L)	2.69–3.12	3.39	3.23	2.87	0.440	0.420	0.957
P (mmol/L)	0.77–1.39	1.84	2.13	1.03	0.866	0.326	0.446
Fe (μmol/L)	13.1–24.7	43.4	36.9	28.5	10.52	0.133	0.570
CREA (μmol/L)	79.6–132.0	57.2	69.4	71.4	19.0	0.278	0.887
Glu (mmol/L)	3.89–5.28	2.47	3.06	3.34	31.7	0.805	0.747
TG (mmol/L)	0.24–1.14	0.99	1.06	0.85	0.39	0.813	0.635
TC (mmol/L)	1.42–2.30	2.18	2.07	1.95	0.40	0.776	0.885

^1^ Gestation: 4-mo. = 4th month of gestation; 7-mo. = 7th month of gestation; 10-mo. = 10th month of gestation.^2^ Item: TP = total protein; ALT = alanine aminotransferase; AST = aspartate aminotransferase; γ-GT = gamma alkaline phosphatase; TB = total bilirubin; CREA = creatinine GLU = glucose; TG = total triglycerides; TC = total cholesterol. ^3^ Reference value: according to Caldin et al., 2005 [8].^4^
*p*-value: *p* < 0.05 indicates significant effect of linear and/or quadratic contrasts.

**Table 3 animals-11-02646-t003:** Correlation coefficients (ρ) and statistical significance (*p*-value in italics below ρ) of selected metabolic parameters at Pearson’s test.

Item ^1^	ALB	TP	AST	ALT	TB	UREA
TP	0.445					
	*0.020*					
AST	0.618	0.595				
	*0.001*	*0.001*				
ALT	−0.075	0.296	0.249			
	*0.709*	*0.134*	*0.211*			
TB	−0.112	−0.248	−0.145	−0.150		
	*0.586*	*0.221*	*0.480*	*0.465*		
UREA	0.283	0.579	0.418	0.293	−0.709	
	*0.153*	*0.002*	*0.030*	*0.138*	*0.000*	
CREA	−0.249	−0.540	−0.362	−0.703	0.206	−0.560
	*0.219*	*0.004*	*0.069*	*0.000*	*0.312*	*0.000*

^1^ Item: TP = total protein; AST = aspartate amino transferase; ALT = alanine amino transferase; TB = total bilirubine; ALB = albumin. CREA = Creatinine.

**Table 4 animals-11-02646-t004:** Concentration of SPF of pregnant jennies at 4th, 7th, and 10th month of gestation.

	Phase ^1^			
Item ^2^	4-mo.	7-mo.	10-mo.	*p*-Value ^3^
ALB (g/L)	26.5 ± 2.45 ^a^	22.3 ± 1.36 ^b^	21.9 ± 0.24 ^b^	0.001
α_1_—Globulins (g/L)	16.1 ± 7.33 ^a^	12.0 ± 1.51 ^ab^	11.3 ± 1.09 ^b^	0.047
α_2_—Globulins (g/L)	14.6 ± 7.02	14.0 ± 9.00	13.7 ± 6.08	0.052
β—Globulins (g/L)	16.4 ± 9.98	16.2 ± 2.14	16.5 ± 1.87	0.954
γ—Globulins (g/L)	8.01 ± 0.19 ^a^	5.57 ± 0.78 ^b^	4.00 ± 0.91 ^b^	0.037
A/G	0.48 ± 0.13	0.47 ± 0.07	0.48 ± 0.04	0.916

^1^ Phase: 4-mo. = 4th month of gestation; 7-mo. = 7th month of gestation; 10-mo. = 10th month of gestation. ^2^ Item: ALB = albumin; A/G = albumin to globulins ratio. ^3^
*p*-value: *p* < 0.05 indicates significant effect of linear and/or quadratic contrasts. ^a, b^ superscripts in a same row indicate a significant difference (*p* < 0.05).

## Data Availability

The data presented in this study are available on request from the corresponding author. The data are not publicly available due to the fact that only raw data are available. All essential information of the article appear in the Tables of the paper.

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
