# Peer review of "Shift in Circulating Serum Protein Fraction (SPF) Levels of Pregnant Jennies and Nutritional Related Aspects at Early-, Mid- and Late Gestation"

_animals, 2021, doi:10.3390/ani11092646_

Round 1

Reviewer 1 Report

My suggestions are:

  • a revision of English language and grammar along the manuscript;
  • to start the introduction with the gestational metaboli aspects, since the nutrition and feeding is a must have to live and to be pregnant, moreover all the animals enrolled in this study were fed on pasture and hay, while the gestational status in jennies is the real topic of this article.
  • the remaining paragraphs are clearly described.

Author Response

Dear Reviewer 1,

many thanks for your review report. All of requests of change were addressed and manuscript amended accordingly. Please, find in attachment a point by point response from authors.

Again, many thanks for your precious expertise and time spent in reviewing our work.

Best regards, 

Maria Grazia Cappai 

for co-authors 

Reviewer 2 Report

Dear Authors,

The novelty of the information that I found in your manuscript needs to be shared with the scientific community. Thus, I highly recommend this paper for publication.

I do feel that the manuscript needs some work on the “English language and style”, but about the scientific value everything checks in my modest opinion.

Because “results” are so finely written, I feel that the “discussion” should match the results in a more refined way. The discussion is sometimes “wordy” and maybe a more step by step approach reprising what previously said in the “results” would allow the reader for a better understanding of the usefulness of the great data that You were able to collect.

Thank you again,

Best regards

Line 17: I feel that the use of “meanwhile” is incorrect. I suggest “Regardless of these basic principles to ensure adequate feeding…”

Line 21: the structure should be “mother-fetus binomial”

Line 22: “In this regard”

Line 28: If using “early-” and “mid-” it should probably be “end-gestation” or “late-gestation” and not “term”. The same is throughout the text.

Line 46-47: “appears to represent” (no “s” at the end)

Line 52: I suggest substituting “acquaintances on digestion physiology” with “Knowledge about the digestive physiology”

Lines 52 to 61: Overall the same concept is repeated and rephrased several times. I suggest rephrasing this first part of the intro.

Line 70: “above all if post absorption” the phrasing makes the concept difficult to understand. I suggest rephrasing.

Line 76: same as before

Line 81: As “SPF” has already been mentioned in the text before, the repeating of its meaning is unnecessary.

Line 83: I suggest “Therefore, we hypothesized…”

Line 94: “animals were handled by”

Line 100-101: I suggest rephrasing “At the same time of successful conception assessment via ultrasound imaging, the nutritional status of all jennies was evaluated following…”

Line 105-106: I suggest rephrasing “The normal gestation period of the Sardo donkey is approximately 350-356 days. Health screening was scheduled following…”

Line 106-109: I suggest rephrasing “Thus, at the fourth, seventh and tenth month from the AI, a physical examination and nutritional assessment was performed on all jennies.”

Line 117-119: I suggest rephrasing. The sentence is too chaotic.

Line 117-122: Please pay attention to the conjugation of verbs.

Line 137: “…directly by jugular vein sampling (18 G needle)”.

Line 226: “mother-fetus binomial”

Line 226-227: “effective tool”

Line 230: “reproductive organs…”

Line 232: “these metabolic…”

Line 238-240: It is difficult to understand what the Authors meant with this sentence. I suggest rephrasing.

Author Response

Dear Reviewer 2,

many thanks for your review report. All of requests of change were addressed and manuscript amended accordingly. Please, find in attachment a point by point response from authors.

Again, many thanks for your precious expertise and time spent in reviewing our work.

Best regards, 

Maria Grazia Cappai 

for co-authors 
